# Proteomic Characterization of SAS Cell-Derived Extracellular Vesicles in Relation to Both BPA and Neutron Irradiation Doses

**DOI:** 10.3390/cells12121562

**Published:** 2023-06-06

**Authors:** Davide Perico, Ying Tong, Lichao Chen, Shoji Imamichi, Yu Sanada, Masamichi Ishiai, Minoru Suzuki, Mitsuko Masutani, Pierluigi Mauri

**Affiliations:** 1Institute of Biomedical Technologies ITB-CNR, Via Fratelli Cervi 93, 20054 Segrate, Italy; davide.perico@itb.cnr.it; 2Department of Molecular and Genomic Biomedicine, Center for Bioinformatics & Molecular Medicine, Graduate School of Biomedical Sciences, Nagasaki University, Nagasaki 852-8523, Japan; bb55320022@ms.nagasaki-u.ac.jp (Y.T.); simamich@ncc.go.jp (S.I.); 3Central Radioisotope Division, National Cancer Center Research Institute, Tokyo 104-0045, Japan; chen202107@outlook.com (L.C.); mishiai@ncc.go.jp (M.I.); 4Institute for Integrated Radiation and Nuclear Science, Kyoto University, Osaka 590-0494, Japan; sanada.yu.6n@kyoto-u.ac.jp (Y.S.); suzuki.minoru.3x@kyoto-u.ac.jp (M.S.); 5Institute of Life Sciences, Sant’Anna School of Advanced Studies, 56127 Pisa, Italy

**Keywords:** BNCT, extracellular vesicles, BPA, proteomics, LC-MS

## Abstract

Boron neutron capture therapy (BNCT) is a selective radiotherapy based on nuclear reaction that occurs when ^10^B atoms accumulated in cancer cells are irradiated by thermal neutrons, triggering a nuclear fission response leading to cell death. Despite its growing importance in cancer treatment, molecular characterization of its effects is still lacking. In this context, proteomics investigation can be useful to study BNCT effect and identify potential biomarkers. Hence, we performed proteomic analysis with nanoLC-MS/MS (liquid chromatography coupled to tandem mass spectrometry) on extracellular vesicles (EVs) isolated from SAS cultures treated or not with ^10^B-boronophenylalanine (BPA) and different doses of neutron irradiation, to study the cellular response related to both boron administration and neutrons action. Despite the interference of fetal bovine serum in the medium, we were able to stratify BPA− and BPA+ conditions and to identify EVs-derived proteins characterizing pathways potentially related to a BNCT effect such as apoptosis, DNA repair and inflammatory response. In particular, KLF11, SERPINA1 and SERPINF2 were up-regulated in BPA+, while POLE and SERPINC1 were up-regulated in BPA−. These results provide the first proteomic investigation of EVs treated with BNCT in different conditions and highlight the potentiality of proteomics for improving biomarkers identification and mechanisms understanding of BNCT.

## 1. Introduction

Boron neutron capture therapy (BNCT) is an anti-tumoral, non-invasive and selective radiotherapeutic technique that uses compounds containing a stable boron isotope ^10^B, typically ^10^B-p-boronophenylalanine (BPA). BNCT is based on nuclear reactions between thermal neutrons and boron-10 atoms (boron neutron capture reaction, BNCR) that are preferentially distributed in cancer cells [1,2,3]. These nuclear reactions release alpha particles and recoil lithium nuclei with short lengths that induce DNA damage, leading to cell death.

The growing importance of BNCT for treating different types of tumors, such as head and neck cancer and gliomas [2], is leading to more and more studies focused on identifying biomarkers for cellular response to BNCT. In particular, proteins related to inflammatory response, DNA repair, RNA processing and apoptosis were typically identified as candidate biomarkers for BNCT response [4,5,6]. Despite that, a molecular characterization with a systems biology approach and high-throughput technologies is still lacking or incomplete [7]. In addition, it is to investigate the effects at the cellular level, due to the single element, boron-containing compounds and neutron irradiation. In this context, proteomics performed with liquid chromatography coupled with high-resolution mass spectrometry (LC-MS) can provide a powerful tool to better understand the molecular mechanisms of cell death induced by irradiation and the selectivity of ^10^B uptake in the tumor site, including the availability to characterize early and late biomarkers to monitor efficacy and side effects and to stratify different samples from a molecular point of view, to potentially describe the environmental context allowing a positive response [8,9,10,11].

Alongside the study of BNCT effect, proteomics and systems biology approaches can also evaluate the effect of boron administration alone (without the radiotherapeutic effect). Boron is present in all organisms, and it is involved in several metabolisms, regulating or interfering with enzyme functionalities [12]. It is important in osteogenesis and bone metabolism [13,14], and it has a role in the inflammatory response and detoxification of ROS [15], so it could lead to an effect in diseases with a strong inflammatory basis, such as cancer. Indeed, it has been reported that the administration of boron-containing compounds can be able to reduce inflammation both in animals and humans, suggesting a beneficial effect of boron as a regulator of inflammatory response and macrophage polarization during immune response [16].

Up to now, there are few investigations on BNCT by proteomics; specifically, the effect due to the single items of therapy, such as boron-containing compounds and neutron irradiation doses, has never been evaluated. Only preliminary data on BPA and BSH (Sodium mercaptododecaborate) effects were allowed to analyze the protein profiles of urine from patients, but these data were not compared to conditions without boron-10 [4].

Among the samples that can be processed and analyzed through proteomic approach, extracellular vesicles (EVs, in terms of microvesicles and exosomes), isolated from cell lines media or biofluids, represent an increasingly used sample for the study of diseases in the field of clinical proteomics, also due to their non-invasiveness. Moreover, they are a critical vehicle for cellular communication, which can be altered in diseases and can contain key signals responsible for cellular alterations, making them a useful source of potential biomarkers [17,18]. In recent years, this has emerged as a fundamental tool for the study and characterization of complex pathologies such as cancer [19] and cardiovascular [20], respiratory [21] and neurodegenerative diseases [22].

Hence, this study aimed to characterize the proteomic profiles of EVs derived from the human tongue squamous cell carcinoma SAS cell line treated or not with BPA, as widely applied in BNCT, and at different times and doses of irradiation with neutrons, in order to study the effect of boron administration and BNCT in different conditions.

In this study, the availability of SAS cells cultured in a serum-conditioned medium containing fetal bovine serum (FBS) limits the depth of proteomic analysis, as bovine serum albumin is the major contaminant of EVs. However, this represents one of the few studies in proteomics applied on BNCT and the first characterizing Evs isolated from samples treated with BNCT, also comparing different conditions (BPA administration and different times of neutron irradiation), and it can provide useful information to improve the understanding of cellular response to boron and BNCT, elucidating mechanisms and the identification of the microenvironment useful for the best response to BNCT treatment.

## 2. Materials and Methods

### 2.1. Neutron Irradiation of SAS Cells

Human squamous cell carcinoma SAS cell line was cultured in DMEM/Ham’s F12 medium containing 10% FBS and 1% penicillin and streptomycin in the humidified CO_2_ incubator with 5% CO_2_ at 37 °C. The ^10^B-Enriched (>98%) BPA was purchased from Katchem spol. s.r.o. (Prague, Czech Republic). The BPA fructose complex was added to the culture medium [6]. Neutron irradiation (0 min, 10 min with 1.9 Gy and 60 min with 11.3 Gy) using 1.5 × 10^6^ cells in 1.5 mL in a vial was carried out in the Kyoto University Research Reactor (KUR), Japan [6]. BPA−fructose complexes at a boron concentration of 25 ppm [^10^B] or mock-control were administrated to the cells for 2 h.

### 2.2. Cell Growth Assay

After neutron beam irradiation in cell suspension, cells were transferred into 6-well plates and cultured for around 8 days. Propagated cells were fixed with 4% (*v*/*v*) formalin solution and stained with 0.1% crystal violet solution as previously described [6]. The area of colonies was measured with Image J software version 1.53a (National Institute of Health, Bethesda, MD, USA).

### 2.3. EVs Isolation from Cell Medium

After neutron beam irradiation in cell suspension, cells were transferred into 6-well plates and cultured for 6 and 24 hrs. The extracellular vesicles (EVs) were isolated from the medium supernatant by centrifugation at 2000× *g* at 4 °C for 20 min, and the supernatant was further centrifuged at 10,000× *g* at 4 °C for 30 min. The supernatant was then centrifuged at 100,000× *g* at 4 °C for 60 min, and the pellet was suspended in PBS and centrifuged again at the same condition, and the pellet was taken. The obtained EVs were kept at −80 °C.

### 2.4. LC-MS/MS Analysis of SAS Cells Medium-Derived EVs

#### 2.4.1. In-Solution Digestion

Proteomic analysis was performed on 3 biological replicates of SAS cells culture media-derived extracellular vesicles (EVs) treated or not with 25 ppm of BPA, with different times and doses of neutron irradiation (0 min with 0 Gy, 10 min with 1.9 Gy and 60 min with 11.3 Gy) and with two-time points of EVs isolation (6 h and 24 h post-irradiation) (*n* = 3, in different conditions). For each EV sample, 100 µg of protein mixture was reduced/alkylated and enzymatically digested using Easy Pep^TM^ Mini MS Sample Prep Kit (Thermo Fisher Scientific, Rockford, IL, USA). Following the kit protocol, in less than 3 h, peptides were generated for each examined condition, cleaned up to prepare detergent-free samples and resuspended in 0.1% formic acid (Sigma-Aldrich Inc., St. Louis, MO, USA) for nLC-hrMS/MS analysis.

#### 2.4.2. Liquid Chromatography

Trypsin digested mixtures were analyzed utilizing a platform consisting of a nano-liquid chromatographic system, the Eksigent nanoLC-Ultra^®^ 2D System (Eksigent, part of AB SCIEX Dublin, Dublin, CA, USA) configured in trap-elute mode, coupled with a high-resolution mass spectrometer. Briefly, samples (0.8 µg injected) were first loaded on a peptide trap (200 µm × 500 µm ChromXP C18-CL, 3 µm, 120 Å) and washed with the loading pump running in isocratic mode with 0.1% formic acid in water for 10 min at a flow of 3 µL/min. The automatic switching of a ten-port valve then eluted the trapped mixture on a nano-reversed phase column (75 µm × 15 cm ChromXP C18-CL, 3 µm, 120 Å) through a 165 min gradient of eluent B (eluent A, 0.1% formic acid in water; eluent B, 0.1% formic acid in acetonitrile) at a flow rate of 300 nL/min. In depth, the gradient was from 5 to 10% B in 3min, from 10 to 30% B in 127 min, from 30 to 95% B in 13 min and holding at 95% B for 9 min.

#### 2.4.3. Mass Spectrometry

MS/MS analyses were performed on an LTQ-OrbitrapXL mass spectrometer (Thermo Fisher Scientific, Monza, Italy) equipped with a nanospray ion source. The spray capillary voltage was set at 1.7 kV, and the ion transfer capillary temperature was held at 220 °C. Full MS spectra were recorded over a 400–1600 m/z range in positive ion mode, with a resolving power of 60,000 (full width at half-maximum) and a scan rate of 2 spectra/s. This step was followed by five low-resolution MS/MS events that were sequentially generated in a data-dependent manner on the top five ions selected from the full MS spectrum (at 35% of collision energy), using dynamic exclusion of 0.5 min for MS/MS analysis. Mass spectrometer scan functions and high-performance liquid chromatography solvent gradients were controlled by the Xcalibur data system version 1.4 (Thermo Fisher Scientific, Monza, Italy).

### 2.5. Proteomic Data Processing and Data Mining

All generated data were searched using the Sequest HT search engine contained in the Thermo Scientific Proteome Discoverer software, version 2.1. The experimental MS/MS spectra were correlated to tryptic peptide sequences by comparison with the theoretical mass spectra obtained by in silico digestion of the UniProt reference proteome. In this case, since the presence of fetal bovine serum in the original samples, both the *Homo sapiens* proteome database (75,550 entries), downloaded on 23 March 2021 (www.uniprot.org), and the *Bos taurus* proteome database (37,511 entries) were used in the SEQUEST HT search engine. The following criteria were used for the identification of peptide sequences and related proteins: trypsin as enzyme, three missed cleavages per peptide, carbamidomethylation of cysteines as fixed modification, methionine oxidation and tyrosine nitrosylation as variable modifications and mass tolerances of ±50 ppm for precursor ions and ±0.8 Da for-fragment ions. Percolator node was used with a target-decoy strategy to give a final false discovery rate (FDR) at Peptide Spectrum Match (PSM) level of 0.01 (strict) based on q-values, considering maximum deltaCN of 0.05 [23]. Only peptides with a minimum peptide length of six amino acids and rank 1 were considered. Protein grouping and strict parsimony principle were applied.

The 72 protein lists obtained from the SEQUEST algorithm were aligned, normalized and label-free compared, excluding bovine identifications. An in-house algorithm, namely, the Multidimensional Algorithm Protein Map (MAProMa), was employed for this aim, using the average peptide spectrum matches (aPSM) [24,25] that correspond to the average of all the spectra identified for a protein and, consequently, to its relative abundance, in each analyzed condition. In depth, to select differentially expressed proteins, subgroups (for BPA− and BPA+ conditions based on the different times and doses of irradiation) were pairwise compared by applying a threshold of 0.2 and 2 on the two MAProMa indexes DAve (Differential Average) and DCI (Differential Confidence Index), respectively. DAve, which evaluates changes in protein expression, was defined as (X − Y)/(X + Y)/0.5, while DCI, which evaluates the confidence of differential expression, was defined as (X + Y) × (X − Y)/2. The X and Y terms represent the PSM of a given protein in two compared samples. In addition, the average protein lists, obtained from each examined condition, were subjected to linear discriminant analysis (LDA), and proteins with the largest F ratio (≥2) and smallest p-value (≤0.05) were retained and processed by hierarchical clustering, applying Ward’s method and the Euclidean’s distance metric using JMP 15.2 software (SAS Institute Inc., Cary, NC, USA). Specifically, the F ratio represented the model mean square divided by the error mean square, whereas the *p*-value indicated the probability of obtaining an F value greater than that calculated if, in reality, there was no difference between the population group means.

The comparison of data versus the Vesiclepedia database [26] was achieved using FunRich v. 3.1.3 (La Trobe University, Bundoora, Australia) [27].

Principal Component Analysis (PCA) was performed using ImageGP (https://www.bic.ac.cn/ImageGP/) [28] accessed on 1 December 2022, with default settings.

Gene Ontology (GO) enrichment analyses were performed using the DAVID database (https://david.ncifcrf.gov, accessed on 1 December 2022). The Select Identifier was set to “UNIPROT ACCESSION”, and the species was set to “Homo sapiens.” Then, the cell component (CC) and biological process (BP) analysis were performed. The data were exported and sorted according to the *p*-value. For each GO enrichment, the items that exceed the thresholds (count: 3 and EASE: 0.001) were selected to draw the advanced bubble diagram by ImageGP [28], using default settings, for BPA+ and BPA− sample groups.

### 2.6. Network Analysis

Starting from the protein list obtained from the combination of LDA and MAProMa results, a protein–protein interaction (PPI) network (93 nodes and 186 edges) was built by the STRING database [29,30]; only experimentally and database-defined PPIs with a score > 0.15 were considered. The resulting sub-network was visualized and analyzed by Cytoscape v. 3.9.1 (National Institute of General Medical Sciences, Bethesda, MD, USA) and its plugins [31]. Proteins were grouped in functional modules with the support of STRING enrichment, using default settings.

## 3. Results

In this study, we evaluated boron-containing compounds and/or the neutron irradiation effect on SAS cells treated with BPA and different doses of neutron irradiation, both at the cellular level, testing cellular growth and viability, and at the molecular level, through proteomics analysis of extracellular vesicles (EVs) derived from cells medium.

### 3.1. Evaluation of Cellular Growth and Viability after BNCT Treatment

After incubation with or in the absence of BPA at 25 ppm and neutron beam irradiation, survival of SAS cells was decreased in cell growth assay condition (Table 1, Figure 1), as previously reported with colony formation assay [6].

### 3.2. Proteomics Profiles of EVs under Different Conditions

The effects of boron and neutron irradiation were investigated by analyzing the protein profiles of the released EVs. Specifically, we performed a differential proteome analysis with a label-free shotgun approach based on nano-liquid chromatography coupled with high-resolution tandem mass spectrometry (nLC-hrMS/MS). Seventy-two runs were acquired by the duplicate analysis of three biological replicates of SAS cell culture medium-derived EVs treated or not with BPA, with different times and doses of neutron irradiation (0 min with 0 Gy, 10 min with 1.9 Gy and 60 min with 11.3 Gy) and with two times of EVs isolation (6 h and 24 h post-irradiation). Excluding bovine proteins identified because of the presence of fetal bovine serum (FBS) in the cell culture media, a total of 2267 distinct human protein groups with at least one unique peptide were identified and mapped to 2198 unique gene symbols, ranging from 1.4 to 4000 kDa for the molecular weight and from 3.7 to 13 for the isoelectric point. For each experimental condition, a unique list was created normalizing and averaging [32] the peptide spectrum match values (aPSM) attributed to the proteins, which represent the number of mass spectra assigned to each and indirectly represent their abundance in the samples (see Appendix A).

A linear regression analysis was performed with software JMP 15.2 to test the technical and biological repeatability of the samples: R^2^ values indicate a good technical and biological repeatability for the samples analyzed, with mean R^2^ of 0.9084, SD = 0.1149 and 0.8871, SD = 0.1513, respectively.

Using FunRich 3.1.3 [27], proteins found with a frequency of at least 2 in the examined conditions were compared to the Vesciclepedia database [26], to assess the origin of identified proteins. As expected, the majority of the identified proteins (72%) have been previously described in EVs and exosomes in the reference database (Appendix A). In particular, we identified EVs markers like tetraspanins (CD9 and CD82), Annexins (ANXA2 and ANXA7) and Rab GTPases (RAB2A, RAB33B, RAB34, and RAB43) that are reported to be found in EVs [26], while we did not identify GM130, which represents a typical marker of cellular contamination [33].

The application of linear discriminant analysis (LDA [34]), on the lists of proteins identified, allowed the extraction of 86 statistically significant proteins (F ratio > 2 and *p* < 0.05), that can discriminate the different groups, to be processed by hierarchical clustering. The cluster analysis (Figure 2a) shows a clear separation in two main branches between BPA− and BPA+ conditions, suggesting a different behavior of EVs following BNCT at early period of 24 h. Moreover, in the BPA+ group is possible to highlight a stratification based on time and dose of irradiation, with a separation between 0, 10 and 60 min, indicating a progressive effect after irradiation, while there was no clear trend in BPA− group in which boron neutron capture reaction does not occur. The last finding was expected due to the absence of the target for neutron irradiation (^10^B).

Similarly, PCA (Figure 2b), performed with the ImageGP platform [28] on the average lists for the different conditions, shows a clear distinction between BPA− and BPA+ conditions along the first component (PC1), while the separation along the second component seems to have had a greater impact on BPA+ conditions, with a similar trend based on post-irradiation time points (6 h and 24 h) for the three different irradiations (0, 1.9 and 11.3 Gy).

The obtained profiles in BPA− and BPA+ conditions were investigated by a functional enrichment analysis to characterize the function of the identified proteins and to evaluate differences in the biological processes involved, possibly related to the BNCT effect. Specifically, Gene Ontology (GO terms) enrichment (Cellular Component and Biological Process) was performed using the DAVID database [35,36], setting filters such as 3 for the count option (minimum number of genes that belong to a certain GO term) and 0.001 for the EASE option (Fisher exact *p*-value) to improve the statistical significance. Enrichment results were plotted using the GO enrichment plot tool of ImageGP, as shown in Figure 3. Despite not having high significance and gene counts, particularly for Biological Processes, at first glance, there is a distinction in the GO terms enrichment between BPA+ and BPA− groups. In particular, based on biological process (Figure 3a), the BPA+ condition presents enriched GO terms that are absent in the BPA− condition, such as DNA repair, apoptotic process, negative regulation of peptidase activity and a higher significance in GO terms like chromatin remodeling, processes potentially involved in inflammation or in a response to radiation damage, while in the BPA− condition there are transcription and cell cycle-related enriched GO terms, compatible with a cancer cell line activity.

### 3.3. Differential Analysis of EVs between BPA+ and BPA− Conditions and within BPA+ Conditions between Different Times and Doses of Neutron Irradiation

The differences between BPA+ and BPA− conditions were investigated to quantitatively examine the proteomic changes, in order to characterize biological pathways related to BNCT effect. Differential analysis was performed comparing the six conditions treated with BPA (three times and doses of neutron irradiation—0, 10 and 60 min—at two time-points of EVs isolation, at 6 h and 24 h post-irradiation) with the equivalent six conditions without BPA treatment. Using the home-made tool MAProMa and applying its two algorithms, DAve (Differential Average) and DCI (Differential Confidence Index), on the aPSMs of every single protein between the two compared terms, it was possible to identify differentially expressed proteins (DEPs) in terms of up- and down-regulated for each considered comparison. MAProMa analysis shows typically a small number of proteins with a differential trend, due to the low quantitative levels for the presence of serum albumin in the samples. Firstly, we examined the differences between BPA+ and BPA− conditions without neutron irradiation (0 Gy) at 6 h and 24 h post-irradiation, to evaluate the potential effect of boron administration through BPA. In the two time points (Figure 4a), there is a substantial balance between up-regulated and down-regulated proteins, and it was possible to highlight some proteins with a common trend: KLF11, SERPINF2 and SRCAP are up-regulated in the BPA+ condition, while SERPINC1, TRAJ56, ARSJ and MCMDC2 are up-regulated in BPA− condition. These proteins are involved in biological functions like apoptosis (KLF11), inflammatory response (SERPINF2 and SERPINC1) and DNA repair (SRCAP and MCMDC2), potentially involved in a cellular response that can be partially determined by the Boron effect even without neutron irradiation.

Other comparisons were focused on the two times of neutron irradiation (10 min with 1.9 Gy and 60 min with 11.3 Gy), both at 6 h and 24 h post-irradiation, always comparing BPA+ with BPA− to evaluate changes due to BNCT effect, also with different dosages to assess the presence of a possible increasing trend (Figure 4b,c). The differential analysis highlights the presence of proteins with a common trend: KLF11, SERPINA1, SERPINF2, ILDR1 and PSD3 are always upregulated in the BPA+ condition, while A2M, POLE, ANKRD12, SERPINC1, SERPINF1 and MCMDC2 are up-regulated in the BPA− condition. For the 1.9 Gy condition, there is a balancing between the two trends, while for the 11.3 Gy condition there is a slightly higher presence of proteins up-regulated in the BPA− group. Most of the extracted proteins maintain the same trend that emerged in the previous comparisons (without irradiation) and the same biological functions involved such as immune response and inflammation (SERPINA1, SERPINF1, SERPINF2, SERPINC1, A2M and ILDR1), apoptosis (KLF11) and DNA repair (POLE and MCMDC2), potentially related to a cellular response to radiotherapy and so to the BNCT effect.

Moreover, to understand if there was a specific trend in proteomic change due to growing time and dose of neutron irradiation (0, 10 and 60 min), MAProMa differential analysis was also performed within the BPA+ group, comparing 0 Gy (without irradiation) with 11.3 Gy (highest irradiation) and 1.9 Gy (lowest irradiation) with 11.3 Gy, both for 6 h and 24 h post-irradiation. The choice of these two comparisons is due to the greater similarity between the profiles of the conditions at 0 min and 10 min compared to that at 60 min, as can be seen from the cluster analysis (Figure 1a). Figure 5 shows the differential analysis of the aforementioned comparisons: in this case, there are no proteins with a common trend in all the comparisons; however, some proteins like SERPINA1, FAM170A, KLF11, HIVEP2 and DHRS7C are typically up-regulated as irradiation increases (11.3 Gy conditions), while ADAMTS10, SRCAP, SERPINF2, ILDR1 and A2M are typically up-regulated in the conditions without irradiation (0 Gy) or with low dose of BNCT (1.9 Gy). Of these, some proteins emerged only in 6 h comparisons (HIVEP2, ILDR1 and SRCAP), while others emerged only in 24 h comparisons (DHRS7C, FAM170A, SERPINF2 and ADAMTS10). Thus, these comparisons show a weaker trend than the results obtained from the differential analysis between BPA+ and BPA− conditions, as hinted by the distances between groups highlighted by cluster analysis. Nevertheless, the presence of a similar group of DEPs, in some cases with similar trends (KLF11 and SERPINA1), suggests that the same biological functions and pathways are involved in the boron effect, BNCT effect and neutron irradiation dose.

Finally, to summarize the most important information about differential trends, the levels of identification in all the different examined conditions of some of the most recurring DEPs through all the comparisons considered are reported in Figure 6. Despite only KLF11 and POLE having good quantitative values, all these proteins exceed MAProMa thresholds in almost every comparison and specific trends can be highlighted. KLF11 and SERPINA1 are always up-regulated in BPA+ conditions and even in correspondence of growing up irradiation time and dose (especially between 0 Gy and 11.3 Gy) and seem to be two of the most involved proteins. POLE and SERPINC1 are typically up-regulated in BPA− conditions, and POLE is slightly up-regulated with lower irradiation. SERPINF2 is absent in BPA− condition, but there is not a clear trend related to the irradiation time change. SRCAP is typically up-regulated in BPA+, except for conditions with higher irradiation (11.3 Gy), which seems to have an inversed regulation.

### 3.4. Network Analysis Shows the Major Pathways Involved in BNCT Effect and How They Are Connected

Combining LDA (86 proteins) and MAProMa tool (27 DEPs) results, 97 proteins were extracted (see Appendix A). Based on this set of proteins, the STRING database [30] was queried to build a network of both known and predicted protein–protein interactions (PPI), visualized then by Cytoscape platform [31]. In the PPI network shown in Figure 7, proteins were represented as nodes grouped in sub-networks based on their molecular function: the most relevant in terms of the number of genes and connections are transcription regulation, innate immune response (typically inflammatory response), cell adhesion, signal transduction, ECM organization, DNA repair and cell cycle, in which there are both statistically significant proteins form LDA and DEPs from MAProMa analysis. Most of these pathways are consistent with a cancer cell line in active proliferation and with a cellular response to radiotherapy (DNA repair, apoptosis and inflammation).

## 4. Discussion

The molecular characterization of complex diseases, especially with high-throughput omics technologies, plays an important role in the definition of new therapeutic strategies and in developing personalized medicine. Here, we presented a first application of proteomic approach of EVs on in vitro study, evaluating singly and combining the two components of BNCT, a boron-containing compound and neutron irradiation. In particular, the analyzed EVs isolated from the cell culture medium represent an important non-invasive source of biomarkers and circulating signals related to cellular alterations and therapeutic effects.

In this study, we used human tongue squamous cell carcinoma (SAS) cells, which represent a type of head and neck cancer that is already treated with BNCT for both in vitro and in vivo experiments [6]. The results of the cell growth assay demonstrated the reduction of cell survival after BNCT treatment (both BPA administration and neutron irradiation), compared with the absence of BPA.

In this case, EVs are isolated from a cell culture medium containing 10% FBS (fetal bovine serum). The use of a conditioned medium with the presence of FBS is necessary for the correct cell growth and survival but determines a problem in the proteomic analysis of secretome and extracellular vesicles due to the interference of highly abundant proteins in serum, such as albumin and immunoglobulins, that can hamper the detection of proteins at low concentrations that are typically the interesting ones [37]. For all these reasons, the presence of FBS is not recommended for proteomic analysis and even depletion of albumin [38]. Although it limits this problem, it is not enough because of possible protein loss. Hence, the best solution may be the serum-free medium, to eliminate all the interferences of bovine proteins, but this is not always possible because of the growth needs of the cell line; this usually does not permit cell growth up to 24 h, and it requires to set up new culture conditions. However, it is possible to exclude bovine proteins in post-processing analysis, trying to overcome these limitations, despite the use of a conditioned serum-free medium would be certainly meaningful [39]. Serum bovine was a shared baseline noise for all considered conditions, and then the identified biomarkers can be confident; of course, bovine albumin hid the identification of proteins expressed at a low level.

Nevertheless, we were able to identify a high number of specific human proteins, with an already described EVs origin, and to discriminate the major conditions, as previously reported with preliminary data on a set of samples [7]. Both cluster analysis and PCA (Figure 2) show that there was a strong separation between BPA− and BPA+ sample groups, indicating a differential response. Moreover, in the BPA+ sample group, the heatmap showed a separation based on time and dose of neutron irradiation, highlighting a progressive trend. This emerged especially for samples irradiated for 60 min, which diverge more from the other two irradiation doses (0 and 1.9 Gy), suggesting a major effect due to prolonged boron-neutron reaction. Instead, PCA highlighted a separation based on post-irradiation time-points (6 h and 24 h), with shared trends in the three different dose conditions, probably due to cell growth effect independently from treatment. In the BPA− sample group, there were not detected clear trends or separations related to doses of irradiation or post-irradiation time points, neither in the heatmap nor in PCA. This finding substantially confirms that there were no significant effects due to neutron irradiation alone without the presence of ^10^B and so without the boron–neutron reaction. Interestingly, samples treated with only BPA (without irradiation) were very close to the ones with both BPA and low doses of neutron irradiation. This can provide insight into a similar effect brought by the presence of boron itself, even without irradiation, supporting evidence on the previously reported anti-cancer effect of boron-containing compounds administration [4,40].

An interesting point to be considered during proteomic data processing is the protein oxidation level. Producing ionizing radiations, BNCT can generate ROS [41], and this underlines the involvement of oxidative stress in BNCT context. Hence, we attempted to preliminary evaluate the oxidation rate of identified proteins considering the modifications that can occur on peptides such as the oxidation of Met and the nitrosylation of Tyr. For both residues, the rate of PSMs for modified peptides compared to total peptides containing Met or Tyr was about 10–11% (767/6671 for Met and 144/1396 for Tyr), and the major contribution in oxidation levels is given by Met (about five-fold). Regarding changes in oxidation levels between the different irradiation conditions during BNCT, we did not detect a difference for Tyr, while for Met there was an increasing trend with higher irradiation (11.3 Gy) (Appendix A), although with low statistically significant difference (*p*-value = 0.15 and 0.09 comparing 0 Gy to 11.3 Gy conditions at 6 h and 24 h, respectively). Despite this preliminary evaluation showing an interesting trend in oxidation levels after BNCT, ad hoc experiments focused on redox proteomics and post-translational modifications research are required [42].

Proteomic characterization allowed the identification of molecular features and biological processes that are differentially regulated in different conditions. Functional enrichment highlighted the presence of different biological functions in the two main sample groups: in the BPA− condition, growth-related terms were detected, such as transcription and cell cycle regulation, indicating an active proliferation, while the BPA+ condition presented terms consistent with a radiotherapeutic effect and a damage response or inflammation, such as apoptosis, DNA repair and chromatin remodeling [43,44,45,46]. These pathways were substantially confirmed by network analysis, including proteins involved in innate immune response, typically with an inflammatory state. For instance, Serpins (Serine protease inhibitors) resulted to play an important role, as several of those were extracted to be differentially expressed proteins. Serpins carry out a variety of biological functions, mainly involved in cellular homeostasis, blood coagulation, immune and inflammatory responses and tissue remodeling [47,48]. Different serpins were identified as biomarkers in cancer, involved in promoting tumor progression and metastasis, suppressing cancer cell viability and regulating inflammation and apoptosis and also related to the specific tumor [49,50,51,52,53]. Among Serpins identified as DEPs, SERPINA1 (α-1-antitrypsin) and SERPINF2 (α-2-antiplasmin) were typically up-regulated in the BPA+ group, while SERPINC1 (Antithrombin-III) was preferentially down-regulated. Of note, none of them were previously reported as BNCT-related biomarkers, although preliminary investigation correlated infusion of boron-containing compounds, as both BPA and BSH, to a differential level of serpins in human urines [4]. In addition, serpins were associated with head and neck cancer; in particular, it has been reported that SERPINA1 is associated with tumor progression [54,55], while SERPINC1 [56] and SERPINF2 [57] are correlated with a good response to radiotherapy, and these data are in good agreement with our results. Here, SERPINA1 increased with the treatment in presence of BPA and even more with neutron irradiation. This is in apparent contradiction with a pro-tumor role already described [54,55], but it was not previously associated with BNCT or radiotherapy on head and neck cancer, and this can be correlated with the inflammatory modulation induced by radiotherapy [46,58].

Among other biological functions involving characterized DEPs, DNA repair was quite present: POLE (DNA polymerase epsilon catalytic subunit A) was down-regulated with neutron irradiation, and SRCAP (Helicase SRCAP) was up-regulated in the BPA+ group except for longer irradiation. These proteins were associated with cancer in the context of double-strand break repair, chromatin remodeling and cell cycle checkpoint regulation [59,60]. Since radiotherapy causes cell death typically via DNA damage and double-strand breaks, irradiated cells activate the signal transduction pathway of DNA damage response (DDR) that mediates DNA repair [61]. However, it has been reported that BNCT can impair DDR by using high linear energy transfer (LET) radiation and thus overcome radioresistance [62], so the initial activation followed by down-regulation of DNA repair proteins with higher irradiation dose can be associated with these observations.

Another interesting protein evidenced in the present study was KLF11 (Krueppel-like factor 11), a transcription factor involved in the induction of apoptosis. This is a typical strategy of cell death induced by irradiation and activated by BNCT over melanomas, gliomas and head and neck cancers [63,64,65]. KLF11 showed an up-regulation in BPA+ condition, with an increasing trend along with neutron irradiation, consistent with previous findings in which radiotherapy induces overexpression of KLF11, suppressing tumor progression through apoptosis activation [66].

Of note, among others identified biological functions, some DEPs with interesting trends were extracted, such as ADAMTS10 (A disintegrin and metalloproteinase with thrombospondin motifs 10), MEGF8 (Multiple epidermal growth factor-like domains protein 8) and ANKRD12 (Ankyrin repeat domain-containing protein 12), that showed differences between BPA+ and BPA− after neutron irradiation. None of these proteins were previously associated with BNCT; however, ANKRD12, a nuclear protein involved in transcription regulation, that resulted down-regulated in the presence of BNCT (Figure 4), was recently reported as a target of a remission-associated miRNA in esophageal squamous cell carcinoma treated with neoadjuvant chemoradiotherapy [67].

As suggested by the trend of differentially expressed proteins through the different conditions, it is possible to speculate that the differences between BPA− and BPA+ groups were more effective in differentiating samples and highlighting biological pathways involved in the cellular response to radiotherapy rather than the comparisons of the different times and doses of neutron irradiation in BPA+ group; thus, the evaluation of the various effects associated with increasing irradiation time/dose and/or the mere administration of BPA was more challenging. However, sample distribution and the regulation of some proteins, especially KLF11, hint that a common regulation may be present with only BPA administration (boron effect) and in combination with neutron irradiation (radiotherapy effect). Therefore, it is conceivable that an initial deregulation induced by the boron effect may be “boosted” by irradiation, although stronger evidence and more exhaustive experiments are required to elucidate this aspect.

Taken together, our results underline the potentiality of proteomic approach to understand the effect of BNCT items (boron-10 and neutron), as well as the importance of using non-invasive samples like EVs; specifically, this study represents a proof of application for investigating BNCT at a molecular level. LC-MS-based proteomics represents a gold standard for protein identification and quantification, because it is highly resolutive, sensitive and reproducible. In our study, the high amount of bovine serum gave an interference that can impair the use of the typical protein validation methodologies such as ELISA and Western Blot, considering also their limitations in quantification, antibodies availability and the number of markers to be investigated. MS-based validation methods, such as Multi-Reaction Monitoring (MRM) [68] or the analysis of a second cohort of samples [69] represent good alternatives, and it would be of great importance to validate proteins in the context of proteomic analysis, although they are time-consuming and require a high number of samples. In the future, it will be necessary to perform other studies on samples, possibly without bovine serum medium, to confirm the reported data with validation methods (especially MS-based) and to increase the knowledge about boron and neutron effects, with the aim to apply this approach also in vivo, on both treated animal models and patients. EVs can be isolated from biofluids (plasma and/or serum) of patients, and proteomics can unveil systemic effects of boron infusion and/or neutron irradiation (BNCT), describing circulating signals correlated with cancer and radiotherapy.

Finally, the continue fast improvements of sensitivity and resolution of proteomic approach can support the clinical practice, including investigation of the sensitivity and efficacy of new boron10- containing compounds and delivery strategies, by means of characterization of target proteins and molecular mechanisms involved in radiotherapy and specifically in boron neutron capture therapy.

## Figures and Tables

**Figure 1 cells-12-01562-f001:**
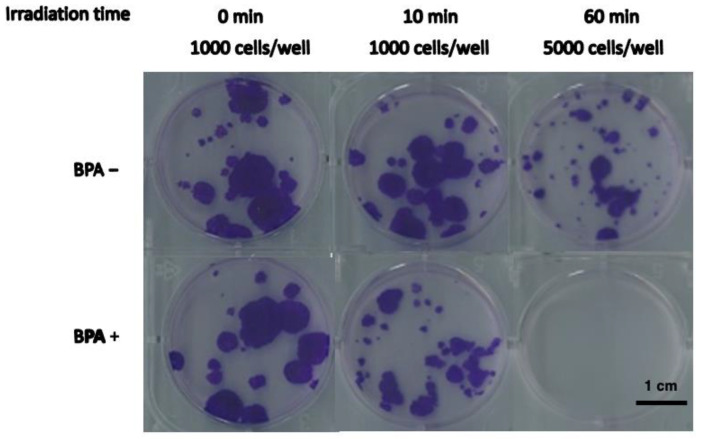
Growth of SAS cells after BNCT or neutron irradiation alone. SAS cells were irradiated with neutrons (BPA−) or as BNCT (pretreatment with BPA at 25 ppm [^10^B]). The cell growth ratio after 8 days culture was analyzed as shown in Table 1.

**Figure 2 cells-12-01562-f002:**
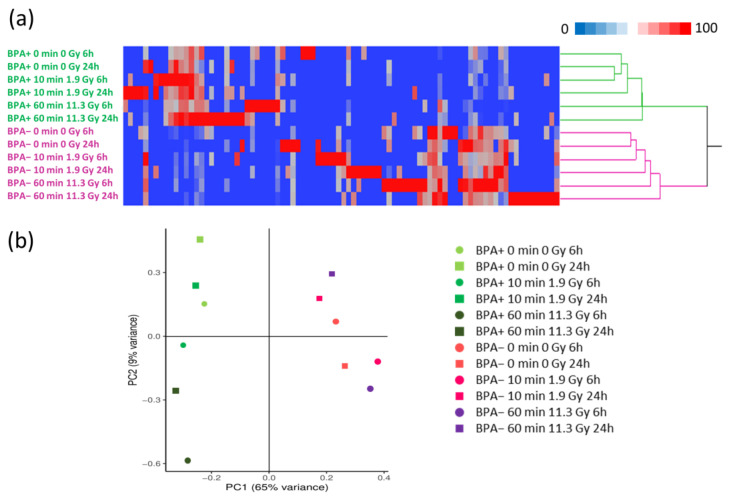
Hierarchical Clustering and PCA of proteomic profiles of EVs derived from SAS cells treated or not with BPA and with different times and doses of neutron irradiation. In Panel (**a**), the dendrogram was obtained by computing the peptide spectrum matches (PSMs) of statistically significant proteins selected by Linear Discriminant Analysis (LDA); Euclidean’s distance metric and Ward ’s methods were applied. In purple are highlighted BPA− samples, while in green are highlighted BPA+ samples. In Panel (**b**), the Principal Component Analysis was performed on the average proteomic profiles for each examined condition. Categories were reported in different colors and shapes according to the represented conditions.

**Figure 3 cells-12-01562-f003:**
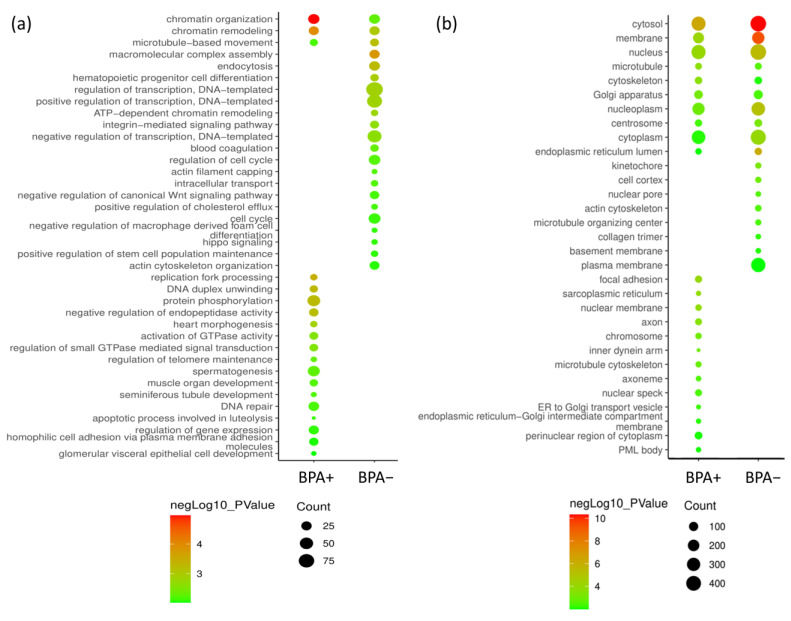
Functional enrichment analysis for BPA+ and BPA− conditions. GO enrichment was performed with DAVID database, and the resulting list of GO terms, filtered for p-value confidence and gene count, was plotted with ImageGP. GO enrichment was performed for biological process (panel (**a**)) and cellular component (panel (**b**)). In plots, the size of circles represents the number of genes associated with a GO term, while the color scale represents the negative Log of p-value as the confidence for the association.

**Figure 4 cells-12-01562-f004:**
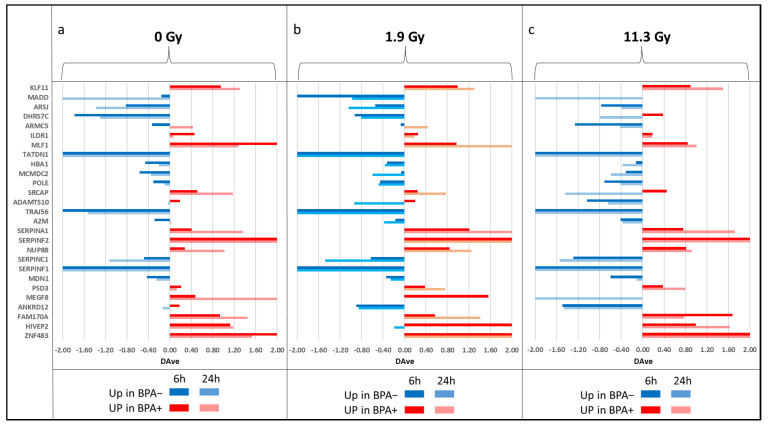
Differentially expressed proteins extracted through label-free quantification with MAProMa tool comparing BPA− and BPA+ conditions with different times and doses of neutron irradiation. Panel (**a**) shows the comparisons without irradiation (0 min, 0 Gy); Panel (**b**) shows the comparisons with low irradiation (10 min, 1.9 Gy), and panel (**c**) shows the comparisons with high irradiation (60 min, 11.3 Gy). In each plot, comparisons at 6 h (darker bars) and 24 h (lighter bars) are reported. Blue bars and negative DAve values refer to upregulated proteins in BPA− conditions, while red bars and positive DAve values refer to upregulated proteins in BPA+ conditions. For each protein, gene name and the related DAve value (ratio of protein expression) are reported. Only proteins with DAve values greater than 0.2 or lower than −0.2 had to be considered DEPs.

**Figure 5 cells-12-01562-f005:**
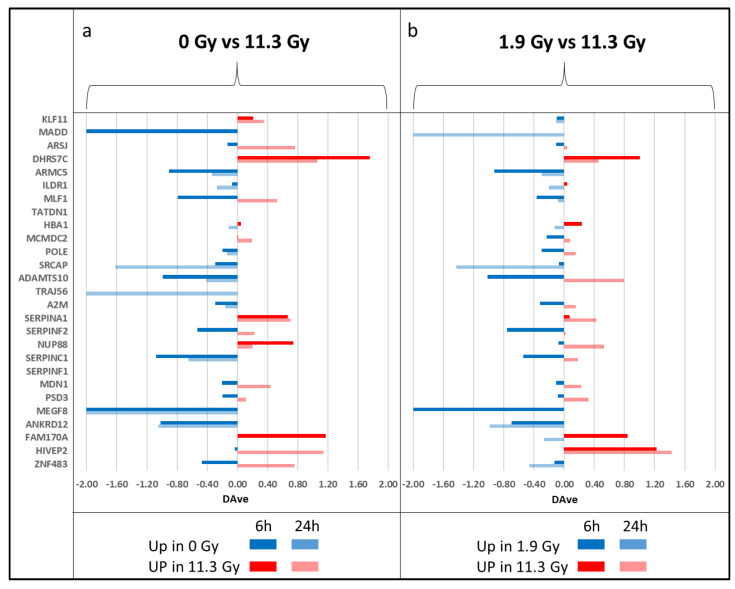
Differentially expressed proteins extracted through label-free quantification with MAProMa tool, comparing conditions of 0 Gy with 11.3 Gy (panel (**a**)) and 1.9 Gy with 11.3 Gy (Panel (**b**)), within BPA+ group. In each plot, comparisons at 6 h (darker bars) and 24 h (lighter bars) are reported. Blue bars and negative DAve values refer to upregulated proteins in conditions with no or low irradiation dose (Panel (**a**,**b**), respectively), while red bars and positive DAve values refer to upregulated proteins in high irradiation dose (11.3 Gy). For each protein, gene name and the related DAve value (ratio of protein expression) are reported. Only proteins with DAve values greater than 0.2 or lower than −0.2 had to be considered DEPs.

**Figure 6 cells-12-01562-f006:**
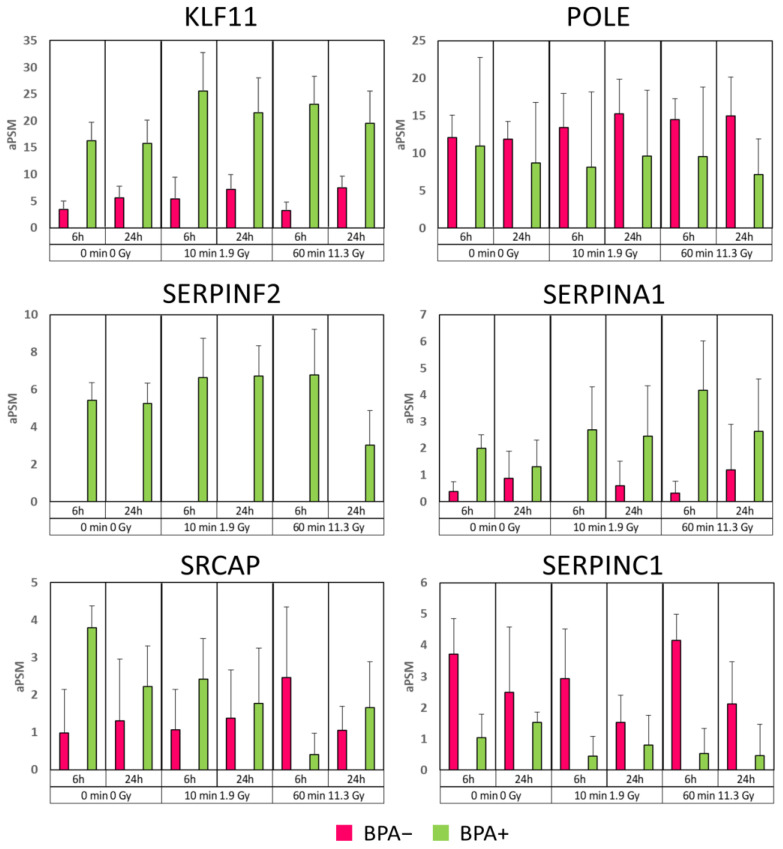
Levels of DEPs through the examined conditions. In each plot, Y-axis represents the average peptide spectrum match (aPSM) of the protein in the different conditions. BPA− and BPA+ conditions are highlighted in different colors (purple for BPA− and green for BPA+). Error bars represent standard deviation.

**Figure 7 cells-12-01562-f007:**
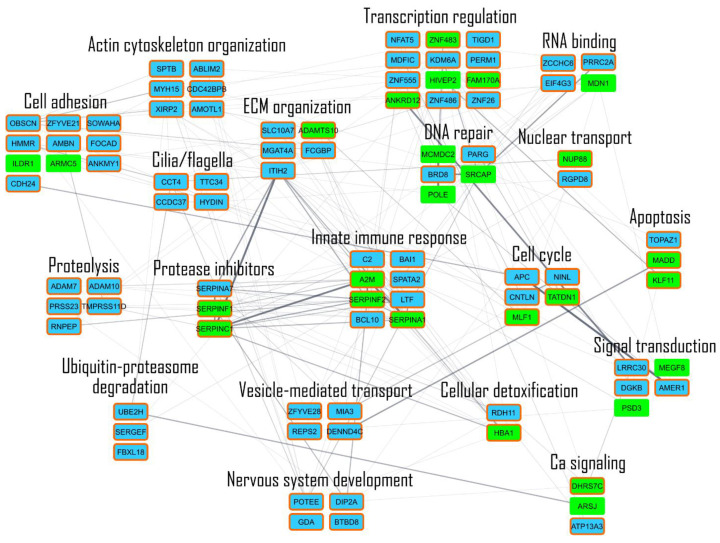
Network analysis. Protein–protein interaction (PPI) network of the proteins obtained combining LDA and MAProMa results. Physical or/and functional interactions are highlighted by thicker edges and considering experimental (STRING score > 0.15) and database (STRING score > 0.35) annotated interactions. The networks were visualized by Cytoscape v.3.9.1 software, while biological processes were retrieved by STRING enrichment. The color code of distinct nodes reflects their sources: orange bound nodes represent proteins extracted from LDA (statistically significant), green nodes represent DEPs and blue nodes represent proteins from LDA that are not DEPs.

**Table 1 cells-12-01562-t001:** The cell growth assay of SAS cells after BNCT treatment.

Irradiated Time (min)	Inoculated Cell Numbers	Cell Growth Ratio *	
		BPA+	BPA−
0	1000	1.00 ± 0.04	1.00 ± 0.11
10	1000	0.40 ± 0.20	1.11 ± 0.15
60	5000	<6.5 × 10^−4^	0.10 ± 0.04

* Mean ± S.E.

## Data Availability

The proteomic datasets (in form of raw data) analyzed for this study can be found in the MassIVE database at the link ftp://massive.ucsd.edu/MSV000091624/ (accessed on 4 April 2023).

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
