# Peer review of "Proteomic Characterization of SAS Cell-Derived Extracellular Vesicles in Relation to Both BPA and Neutron Irradiation Doses"

_cells, 2023, doi:10.3390/cells12121562_

Round 1

Reviewer 1 Report

The authors characterized the proteomic profiles of EVs derived from SAS treated or not with BPA to study the effect of boron administration and BNCT in different conditions. The authors gave sufficient discussion though out the study. However, the manuscript can be further improved by answering the below concerns.

1.       There were rare cells left after 60 min with 11.3 Gy irradiation. It is not clear how the authors determined the dose and irradiation time. Why 10 min with 1.9 Gy and 60 min with 11.3 Gy, but 60 min with 1.9 Gy and 10 min with 11.3 Gy were not compared. Also, to collect 100 ug EVs, usually E10 ~E11 cells are needed. How would the 60 min with 11.3 Gy group have enough cells to secrete EVs?

2.       The author didn’t show any EV markers detected, such as CD81 and PDCD6IP, which cast doubt on the purity of EVs. Density gradient centrifugation or size exclusive column are usually needed to collect purified EVs for proteomics. It is not only albumin, but possible cellular contamination exists, such as GM-130. The authors need to show there is no cellular contamination through proteomic analysis.

3.       It is not clear how to collect EV from cancer patients and how to tell the efficacy of radiotherapy on this patient. The authors should add discussion to show the promising and the meaning of the study.

4.       In Figure 5, some genes are empty, which can be deleted. Such as TATDN1, SERPINF1. The figure should be better arranged, and the genes should be re-ordered.

5.       If possible, some basic characterization of EVs is suggested. For example, NanoSight and TEM.

Reviewer 2 Report

The manuscript describes a pertinent analysis of the effects of boron neutron capture therapy on the EVs proteome. Authors show a high quality proteomic analysis but validation of proteomic results is missing and probably oxidation of amino acid residues was underestimated.

1.       Please make explicit in the text why there was no validation of proteomic data with an alternative methodology like ELISA or Western blot

2.       Ionizing radiation acts on aqueous solutions by destroying water molecules and creating ROS (OH•, O2•- and H2O2). The energy released during the reaction :

10B + nth → [11B] *→ α + 7Li + 2.31 MeV

May be enough to break water molecules. Please add to the discussion section the following points:

a)      Are ROS produced inside EVs and in the cytosol during the boron neutron capture therapy?

b)      If so, What amino acid residues could be oxidized besides Met? (only oxidation of Met was considered in line 160).

c)       If other residues could be oxidized by ROS, How could such variable modification could have affected data processing and detection of differentially regulated proteins?

d)      Did authors detected differences in the amount of oxidized residues when the 11.3 Gy was compared to 0 Gy?

3.       Why authors did not pay attention to the differential regulation of ADAMTS10, MEGF8, ANKRD12?  In Fig. 4 these proteins showed interesting differences between 0 and 11.3 Gy.

Minor points.

Line 245. It must be Figure 2 instead of Figure 1.

Round 2

Reviewer 1 Report

The authors have addressed the questions.

Reviewer 2 Report

The previous concerns have been properly addressed. In my opinion, the manuscript can be accepted.